# A kinase to cytokine explorer to identify molecular regulators and potential therapeutic opportunities

**Marina Chan[1†], Yuqi Kang[1†], Shannon Osborne[2†], Michael Zager[2], Taranjit S Gujral[1,3]***

[1]Human Biology Division, Fred Hutchinson Cancer Center, Seattle, United States; [2]Data Visualization Core, Fred Hutchinson Cancer Center, Seattle, United States; [3]Department of Pharmacology, University of Washington, Seattle, United States

*For correspondence:
tgujral@fredhutch.org

†These authors contributed equally to this work

Competing interest: The authors declare that no competing interests exist.

**Abstract** Cytokines and chemokines are secreted proteins that regulate various biological processes, such as inflammation, immune response, and cell differentiation. Therefore, disruption of signaling pathways involving these proteins has been linked to a range of diseases, including cancer. However, targeting individual cytokines, chemokines, or their receptors is challenging due to their regulatory redundancy and incomplete understanding of their signaling networks. To transform these difficult-to-drug targets into a pharmacologically manageable class, we developed a web-based platform called KinCytE. This platform was designed to link the effects of kinase inhibitors, a well-established class of drugs, with cytokine and chemokine release and signaling networks. The resulting KinCytE platform enables users to investigate protein kinases that regulate specific cytokines or chemokines, generate a ranked list of FDA-approved kinase inhibitors that affect cytokine/chemokine activity, and explore and visualize cytokine signaling network thus facilitating drugging this challenging target class. KinCytE is freely accessible via https://atlas.fredhutch.org/kincyte.

## eLife assessment

This manuscript describes an **important** web resource for kinases connected to cytokines. The **compelling** information will be used by researchers across a number of fields including analysts, modelers, wet lab experimentalists and clinician-researchers, who are looking to improve our understanding of pathologies and means to correct them through modulating the immune response.

## Introduction

Cytokines and chemokines are a group of proteins that are secreted by cells and play a fundamental role in cellular communication (*Fajgenbaum and June, 2020*). These molecules can function through autocrine or paracrine signaling mechanisms, and are responsible for a wide range of biological processes, including inflammation, immune response, cell proliferation, differentiation, and survival (*Becher et al., 2017*). However, when the regulation of these signaling pathways is disrupted, it can lead to the development of several disorders, including autoimmune diseases, allergies, chronic inflammation, and cancer (*Fajgenbaum and June, 2020*). Inflammatory cytokines, such as interleukin-1 (IL-1), interleukin-6 (IL-6), and tumor necrosis factor-alpha (TNF-α), play a significant role in the pathogenesis of infectious diseases. Additionally, chemokines such as CXCL8 are involved in the recruitment of immune cells to the site of infection or inflammation. In the context of cancer, cytokines and chemokines are involved in the initiation, progression, and metastasis of tumors (*Lippitz and Harris, 2016*). Tumor cells can produce cytokines and chemokines to evade the immune system and

promote angiogenesis, whereas disruption of cytokine and chemokine signaling pathways in cancer cells can lead to resistance to chemotherapy and radiation therapy (*Jones et al., 2016*). Thus, identifying the molecular regulators and drugs that could inhibit cytokine and chemokine release could result in the development of novel treatments for various disorders.

Delineating the regulation of cytokines and chemokines pose a challenge due to their functional redundancy (*Chauhan et al., 2021*). Several cytokines may bind to identical receptors and prompt comparable or diverse downstream functions. Moreover, the intricacies of the downstream molecular players that are components of cytokine/chemokine signaling networks remain incompletely understood. As a result, directly targeting a single cytokine or cytokine receptor has been a challenging task. Here, we present KinCytE, a kinase-to-cytokine explorer, a powerful new platform that was designed to allow researchers to explore the downstream molecular regulators and potential drugs that could inhibit the function of one or more cytokines. We anticipate that KinCytE will be highly sought after by biologists from various backgrounds, including immunologists, cancer biologists, virologists, and beyond.

## Results and discussion
### Measuring and modeling cytokine and chemokine responses in human macrophages

To model cytokine and chemokine release and responses, we used human macrophages stimulated with lipopolysaccharide (LPS) treatment, a widely used method for activating macrophages and inducing cytokine and chemokine secretion in vitro and in vivo (*Hume, 2015*). Therefore, we exposed human monocyte-derived macrophages to LPS and analyzed 191 secreted factors (chemokines, cytokines, and growth factors) in conditioned media using a recently developed nano-ELISA (nELISA) method (*Dagher et al., 2023*). Out of the 191 secreted factors analyzed, we observed reliable signals for 37 proteins (*Supplementary file 1*). Our results indicated significant changes in 34 cytokines, chemokines, and secreted factors following 24 hr of stimulation with LPS. These changes included upregulation of previously known cytokines and chemokines such as IL-1, IL-6, IL-10, CCl3, CCL7, and G-CSF (*Figure 1A*). Furthermore, we also found significant upregulation of VEGFa and MMP1 (*Figure 1A*). In addition, we observed significant downregulation of CCL24 (IL-4), a negative regulator of the LPS signaling (*Bonder et al., 1999*). Therefore, the nELISA assay allowed us to identify a wide range of chemokines and cytokines that are released in response to macrophage activation.

To identify signaling pathways activated in macrophages in response to LPS stimulation and explore kinase inhibitors that could inhibit these pathways, we employed a recently developed strategy called KiR (*Bello et al., 2021b*; *Chan et al., 2021*; *Gujral et al., 2014*). The KiR approach uses large-scale drug-target profiling data and broadly selective chemical tool compounds to pinpoint specific kinases underlying a given phenotype (*Rata et al., 2020*). For the purposes of KinCytE development we focused on the release of secreted factors, e.g., cytokines as the phenotype, and quantified the effects of 34 computationally chosen kinase inhibitors on changes in LPS-mediated release of 37 secreted factors (*Figure 1C*, *Supplementary file 2*). Using this training dataset, we built elastic net regularization models to predict kinases essential for each of the 37 secreted factors. We evaluated model performance using leave-one-out cross-validation (LOOCV) mean squared error (MSE) between predicted and observed drug response. In LOOCV, each time 36 drugs' activity profiles were used to train the model to predict the remaining drug's effect on LPS-mediated cytokine release, and MSE between predicted and observed cytokine levels was used to assign an error score to each model (*Figure 1—figure supplement 1*). In total, 37 models successfully met the criteria for optimization, with the requirement of a normalized root MSE of less than 0.1 and a Pearson correlation exceeding 0.85 when comparing predicted and measured responses. These 37 models collectively identified 168 'informative kinases' from a pool of 298 kinases, suggesting their potential involvement in the LPS-mediated release of one or more factors (*Figure 1D*, *Supplementary file 3*). These include several kinases known to play a critical role in macrophage survival and activation and cytokine signaling, such TAK1 (*Irie et al., 2000*), Src Family kinases (*Smolinska et al., 2008*), JAK1, IRAK, GSK3 (*Noori et al., 2020*), FES (*Parsons and Greer, 2006*), p38 MAPK and CK2 (*Glushkova et al., 2018*). Among these kinases, Src family kinases, GSK3, CK2, and TAK1 were predicted to be important for the release of more than 10 cytokines and chemokines. In addition, our model highlighted the importance of various

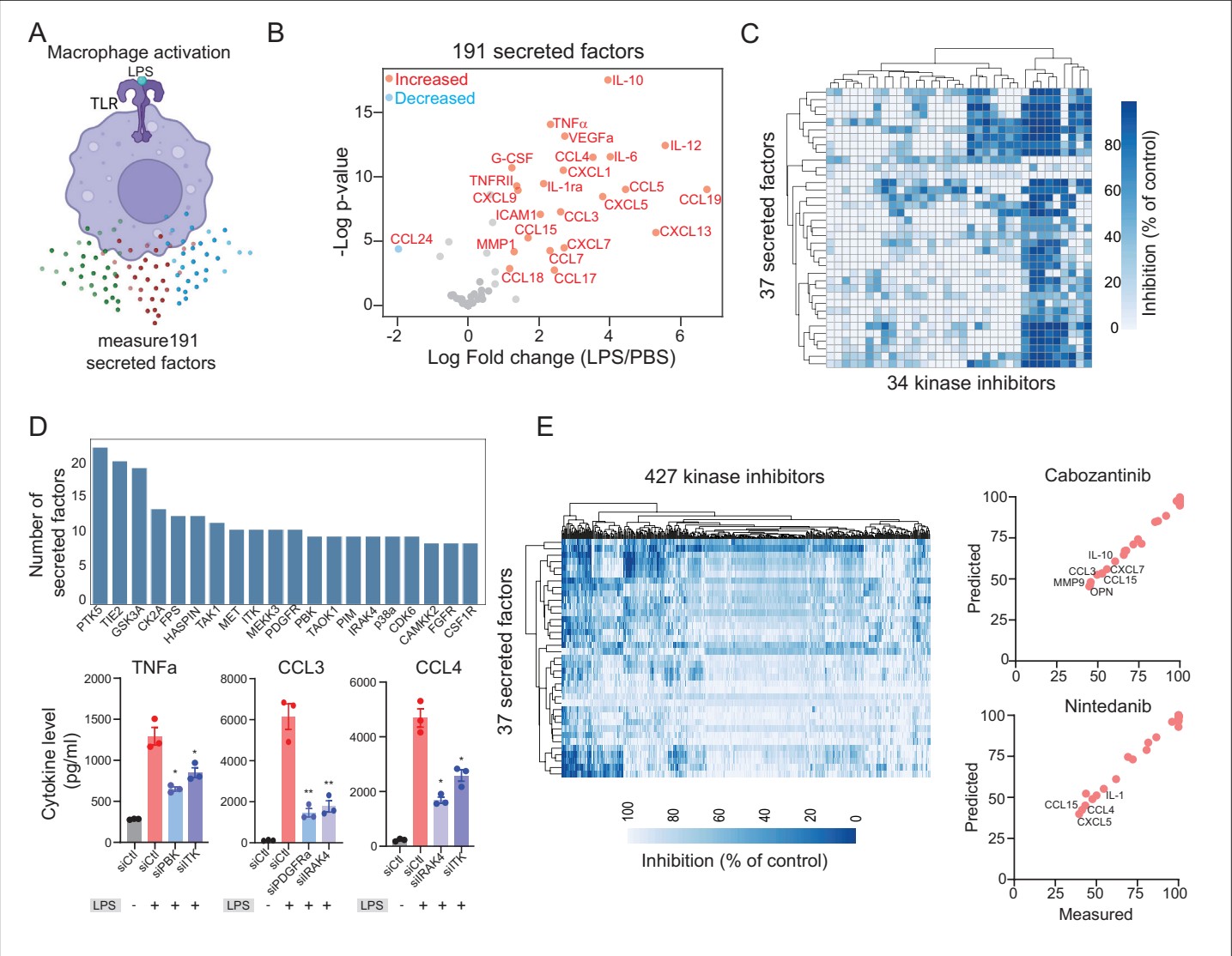

**Figure 1.** Modeling lipopolysaccharide (LPS)-mediated changes in secreted factors in human macrophages. (**A**) A schematic showing LPS-mediated activation of human macrophages. (**B**) A plot showing changes in the levels of 191 secreted factors in response to LPS stimulation. Significantly increased or decreased factors are also indicated. (**C**) A heatmap showing changes in LPS-mediated 37 secreted factors in response to 34 kinase inhibitors. (**D**) Key kinases that regulate cytokines and chemokines. Top, a plot showing the top kinase and the number of secreted factors they regulate. Bottom, experimental validation of change in TNF-α, CCL3, and CCL4 upon knockdown of indicated kinases. (**E**) A heatmap showing predicted changes in 37 secreted factors in response to 427 kinase inhibitors. Right, experimental validation showing correlation between measured and predicted changes in 37 cytokines in macrophages treated with FDA-approved cabozantinib and nintedanib at 500 nM.

The online version of this article includes the following figure supplement(s) for figure 1:

**Figure supplement 1.** Representative KiR models for OPN and CCL4 release.

receptor tyrosine kinases including FGFR, CSF1R, VEGFR, and MET. We used small interfering RNA (siRNA) knockdown experiments to validate involvement of specific kinases in regulation of three main cytokines. We observed that in all cases, knockdown of a kinase leads to a decrease in cytokine release (*Figure 1D*). Moreover, our analysis suggested that a single kinase may play a vital role in the release of several cytokines, and conversely, a given cytokine may be regulated by multiple kinases, indicating a complex interplay between kinases and cytokines.

Given that our analysis implicated multiple kinases in affecting cytokine release, we sought to identify poly-specific kinase inhibitors that could target the release of one or more cytokines and chemokines. To achieve this, we used the optimized KiR models to predict single-agent responses to 427 kinase inhibitors (*Figure 1E*, *Supplementary file 4*). Our kinase inhibitor collection includes

39 FDA-approved kinase inhibitors and numerous compounds currently undergoing clinical testing. Among these, ponatinib, abemaciclib, and nilotinib were the top hits predicted to inhibit the release of over 15 cytokines and chemokines (*Supplementary file 4*). Our findings are consistent with a previous study demonstrating the inhibitory effect of ponatinib on LPS-mediated macrophage activation both in vitro and in a mouse model of lung inflammation (*Chan et al., 2021*). Additionally, we experimentally confirmed the efficacy of three inhibitors, cabozantinib, nintedanib, and dovitinib, and observed an accuracy of over 80% in line with the KiR model's predictions (*Figure 1E*). Consequently, our systems-based modeling efforts enabled us to predict the efficacy of a vast number of kinase inhibitors in inhibiting the activity of one or more secreted factors. These results hold promise for repurposing existing inhibitors to treat diseases where cytokine release plays a significant role, such as sepsis and viral-mediated infections.

Finally, protein kinases identified through KiR can be used to deduce signaling pathways that are involved in a specific biological process. We have recently developed a new computational approach known as KiRNet, which allows construction of signaling networks based on the KiR (*Bello et al., 2021a*). We built kinase-centric networks for each of the 31 cytokines, chemokines, and other secreted factors by applying pathway analysis tools, including KEGG (*Kanehisa and Goto, 2000*). These networks can be accessed and explored using the KinCytE tool (see below). Overall, by employing a combination of multiplex measurements, functional screening, and computational methods, we were able to achieve two key outcomes: (1) the creation of network-centric portraits that govern the release of particular cytokines and chemokines, and (2) the generation of ranked lists of kinase inhibitors, including some that contain FDA-approved drugs, that highlight the potential of known drugs to inhibit the function of one or more secreted factors.

## Development and use of KinCytE

We designed KinCytE as a publicly accessible online platform that could be used to address fundamental questions in cellular biology, such as 'Which signaling pathways are downstream of cytokine and chemokines?', 'Which compounds are suitable for inhibiting a specific cytokine or chemokine pathway?', and similar. To begin answering these questions, the first and often most crucial step is to systematically determine the changes in a large number of chemokines and cytokines in response to various perturbations. Leveraging the primary dataset on LPS-induced changes in secreted factors and KiR model-predicted kinases and response to kinase inhibitors, we created KinCytE as a tool for exploring cytokine and chemokine signaling. KinCytE is an intuitive web-based platform that enables users to rapidly determine the essential kinases responsible for releasing particular cytokines or groups of cytokines. Additionally, the platform offers a ranked list of currently available FDA-approved and other drugs that have the potential to inhibit cytokine/chemokine activity. Finally, the user-friendly graphical user interface facilitates visualization and exploration of cytokine signaling network maps.

The KinCytE app runs entirely in-browser and can be found at https://atlas.fredhutch.org/kincyte/. Upon loading the app webpage, users can: (1) utilize this app to identify drugs that potentially reduce the level of cytokines of interest (COI) via clicking on 'Identify Drugs', which will bring them 'Drug Discovery' page; (2) view kinases that could be involved in the regulation of the COI via clicking on 'Identify Kinases', which would bring them to 'Kinase Discovery' page; and (3) explore the connection between COI and its related kinases via clicking on 'Explore Cytokines', which leads to 'Cytokine Explorer' page (*Figure 2A*). Each page provides instruction about the expected content and format of input and output.

Kinase Discovery page gives a matrix of kinases that might regulate COI (*Figure 2B*). The table assigns '1' to a kinase-cytokine combination where kinase could regulate the cytokine and '0' otherwise. Rows of COI are pinned at the top panel. This matrix also summarizes the number of cytokines within COI and among all cytokines that a kinase could affect, with details of its impact on other cytokines that are not included in COI.

Drug Discovery page generates a bar chart and a table based on the given cytokine(s) (*Figure 2C*). The bar chart shows top 10 drugs with highest average efficacy, which is defined by the percentage of the cytokine/chemokine level change compared to the control level. Deep blue color refers to drugs that affect all cytokines while light blue color indicates drugs with narrower scope of targets. Users hover on each bar to get the exact number. The table presents the summary of all 427 drugs from following aspects: average efficacy on COI, percentage of COI affected, drug selectivity and

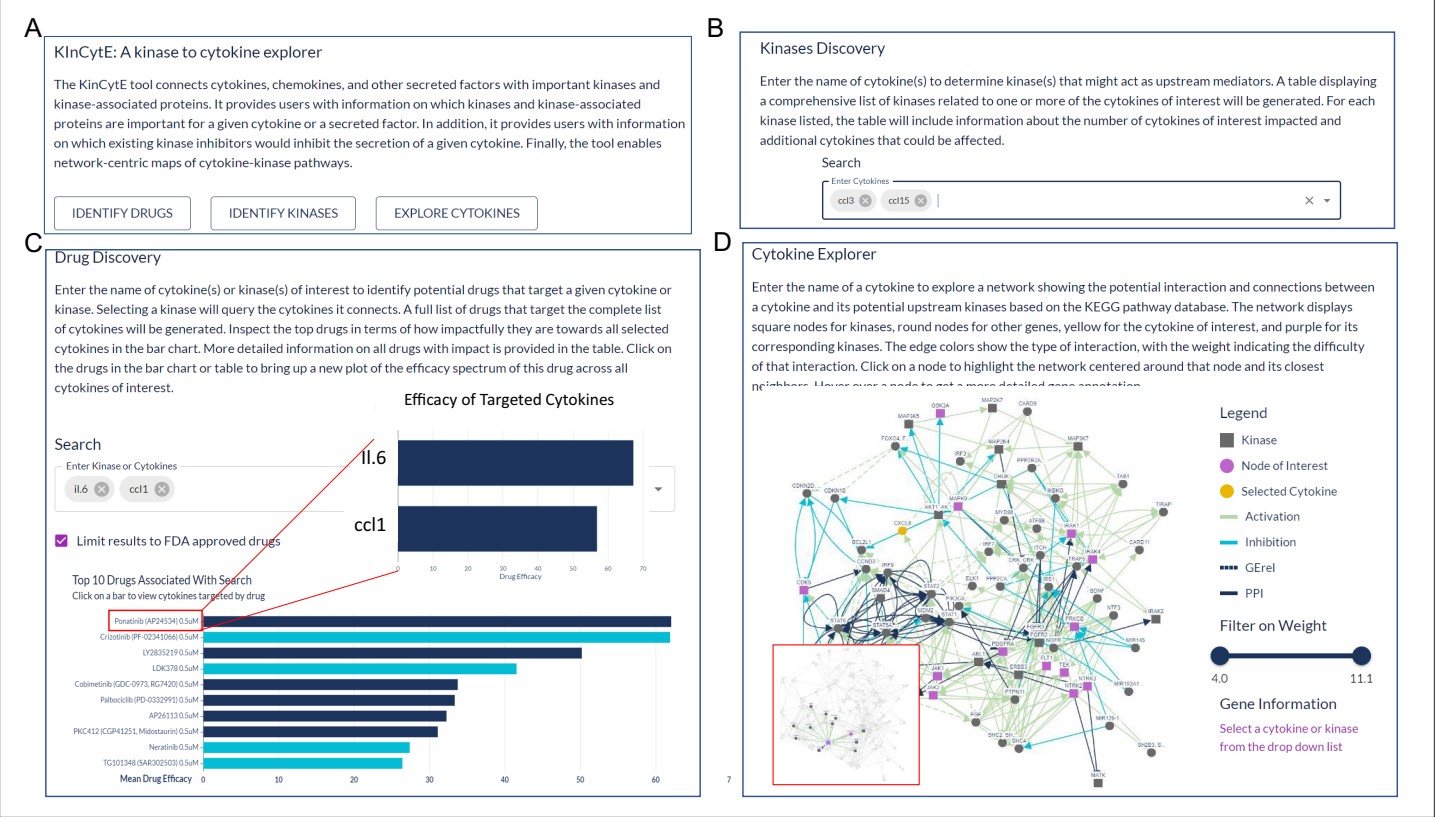

**Figure 2.** An illustration showing the steps for using KinCytE. (**A**) KinCytE offers three sections for users to explore potential cytokine(s) or chemokine(s) inhibiting drugs (Identify Drugs), related kinases (Identify Kinase), and cytokine signaling network (Explore Cytokines). Users can click on any of these sections to access the desired information. (**B**) Users can obtain information on kinase(s) relevant to the expression of selected cytokines by selecting the desired cytokine(s) from the available options. This will generate a table with the relevant information. (**C**) Users select cytokine(s) to view a bar plot showing the top 10 most potent drugs in inhibiting the expression of these cytokine(s), with a table recording the effectiveness of all 427 drugs. Users can click on the drug name in either the table or the bar plot to access a more detailed cytokine efficacy profile of that drug. (**D**) To produce a network showing the connection between a selected cytokine and its related kinases, users need to select the desired cytokine. This will generate a network with edges representing the connections between the cytokine and its related kinases. Users can slide the weight bar to prune edges and click on a node to get the annotation of the gene(s) represented by that node. All tables in KinCytE are fully sortable, filterable, and downloadable for the convenience of users.

specificity, and FDA approval status. Users clicks drug name on either bar chart or table for a new plot showing the efficacy profile of this drug on COI.

Cytokine Explorer page produces a graphic connection between COI and its potential upstream kinases based on the KEGG analysis (*Figure 2D*). Node represents gene(s), and edge represents interaction between the two nodes (genes) it connects. More detailed annotation is provided in the legend of each network figure. Users slide the bar to filter out edges of higher weight, which indicates less likely interaction. Users click on a node to drag it, highlight its neighboring network, and obtain additional biological information of gene that this node refers to.

KinCytE links the release of cytokines and chemokines to kinase activity and recommends drugs that target specific cytokine sets, as well as identifies kinases and kinase-related proteins that may be involved. This platform also extends the kinase-cytokine relationship by incorporating information from biological pathways. Its architecture and standardized data processing facilitate effortless updates to incorporate new cytokines and other immune factors with corresponding experimental data. We welcome researchers to actively participate in advancing the development of KinCytE by sharing external screening data, especially data on new secreted factors and cell types that extend beyond macrophages. This collaborative effort promises to enhance our understanding of kinase-focused networks, opening new avenues for cutting-edge therapeutic approaches.

# Materials and methods

**Key resources table**

| Reagent type (species) or resource | Designation | Source or reference | Identifiers | Additional information |
|---|---|---|---|---|
| Biological sample (*Homo sapiens*) | Human CD14+ monocytes | Bloodworks Northwest | Cat # 4585-18 | |
| Chemical compound, drug | Kinase inhibitors used for KiR model screening | National Center for Advancing Translational Science, NIH | See **Supplementary file 2** | |
| Software, algorithm | R 4.2.1 | *Venables and Smith, 2003* | https://www.r-project.org/ | |
| Software, algorithm | glmnet | *Friedman et al., 2021* | https://cran.r-project.org/web/packages/glmnet/index.html | |
| Software, algorithm | ggrepel | *Slowikowski et al., 2018* | https://github.com/slowkow/ggrepel | |
| Software, algorithm | tidyverse | *Wickham et al., 2019* | https://tidyverse.tidyverse.org/index.html | |
| Software, algorithm | ggpubr | *Kassambara, 2023* | https://rpkgs.datanovia.com/ggpubr/index.html | |
| Software, algorithm | ggplot2 | *Wickham, 2009* | http://ggplot2.org | |
| Software, algorithm | pheatmap | *Kolde and Kolde, 2015* | https://cran.r-project.org/package=pheatmap | |
| Software, algorithm | Python 3.7.11 | *Van Rossum and Drake, 2009* | https://www.python.org/downloads/release/python-3711/ | |
| Software, algorithm | Numpy | *Harris et al., 2020* | https://numpy.org/ | |
| Software, algorithm | Pandas | *McKinney, 2010* | https://pandas.pydata.org/ | |
| Software, algorithm | Scipy | *Virtanen et al., 2020* | https://scipy.org/ | |
| Software, algorithm | Seaborn | *Waskom, 2021* | https://seaborn.pydata.org/ | |
| Software, algorithm | Matplotlib | *Hunter, 2007* | https://matplotlib.org/ | |

## Cell lines

Human CD14+ monocytes cells were obtained from Bloodworks Northwest. Cells were grown at 37°C under 5% $CO_2$, 95% ambient atmosphere and maintained in Roswell Park Memorial Institute (RPMI) 1640 Medium supplemented with 10% FBS (Sigma) and 1% Penn Strep. Macrophages were derived from CD14+ monocytes using M-CSF Recombinant human protein at a final concentration of 50 ng/mL for 6 days.

## Quantitative profiling of the secretome

Conditioned media from macrophages treated with LPS for 24 hr were collected and sent to NomicBio (Montreal, Canada) for nELISA-based cytokine analysis as described previously (*Dagher et al., 2023*).

## Kinase inhibitor screening

Kinase inhibitor screening was performed as described previously (*Chan et al., 2021*). Monocyte-derived macrophages were plated in 96-well plates. Next days, cells were treated with LPS and a panel of 34 kinase inhibitors at 500 nM doses in duplicate. Conditioned media was collected after 24 hr for the secretome analysis as described above.

## KiR modeling

Elastic net regularized models, KiR models, and the list of predicted key kinases or key functional nodes were generated as previously described (*Bello et al., 2021a*; *Chan et al., 2021*; *Gujral et al., 2014*). Briefly, a panel of 427 kinase inhibitors previously had their pairwise effects on 298 human kinases profiled. The result is a quantitative drug-target matrix, where each entry is a percentage between 0 and 100 that represents that kinases' residual activity (as a percent of control, uninhibited

activity) in the presence of that inhibitor. The kinase inhibition profiles of each inhibitor and the quantitative responses to those inhibitors were used as the explanatory and response variables, respectively, for elastic net regularized multiple linear regression models. Custom R scripts (available at https://github.com/FredHutch/KiRNet-Public, copy archived at *FredHutch, 2020*) employing the glmnet package were used to generate the final models. LOOCV was used to select the optimal value for the penalty scaling factor 1. Models were computed for 11 evenly spaced values of a (the relative weighting between LASSO and Ridge regularization) ranging from 0 to 1.0 inclusive. Kinases with positive coefficients in at least one of these models (except for a=0, which always has non-zero coefficients for every kinase) were considered hits. Model accuracy was assessed via the LOOCV error as well as the root MSE of the predictions for the tested inhibitors.

### siRNA transfection

All siRNA were obtained from Dharmacon (Thermo). siRNA transfections in 12-well plate for measuring changes in secreted factors were carried out using Lipofectamine RNAiMax (Invitrogen) according to the manufacturer's instructions.

### Quantification and statistical analysis

All software development, calculations, and analyses were carried out using R 3.3.0 (https://www.r-project.org/). All packages used can be found in the Key Resources Table.

## Acknowledgements

This study was supported by grant from the National Science Foundation (2047289).

## Additional information

### Funding

| Funder | Grant reference number | Author |
| --- | --- | --- |
| National Science Foundation | 2047289 | Taranjit S Gujral |

The funders had no role in study design, data collection and interpretation, or the decision to submit the work for publication.

### Author contributions

Marina Chan, Data curation, Validation, Investigation, Methodology, Writing - review and editing; Yuqi Kang, Data curation, Formal analysis, Visualization, Methodology, Writing - original draft; Shannon Osborne, Software, Visualization, Methodology; Michael Zager, Software, Visualization, Project administration; Taranjit S Gujral, Conceptualization, Data curation, Supervision, Funding acquisition, Methodology, Writing - original draft

### Author ORCIDs

Michael Zager http://orcid.org/0000-0002-9416-8685
Taranjit S Gujral http://orcid.org/0000-0002-4453-3031

Reviewer #1 (Public Review): https://doi.org/10.7554/eLife.91472.3.sa1
Reviewer #2 (Public Review): https://doi.org/10.7554/eLife.91472.3.sa2
Author Response https://doi.org/10.7554/eLife.91472.3.sa3

## Additional files

### Supplementary files
• MDAR checklist
• Supplementary file 1. Fold change of average cytokine level in LPS-treated macrophage compared

to control.

- Supplementary file 2. Percentage changes in cytokine level in kinase inhibitor-treated macrophages compared to control.
- Supplementary file 3. Summary table of predicted kinases involved in cytokine models.
- Supplementary file 4. Predicted cytokine levels in macrophares treated with kinase inhibitors.

### Data availability

The app portal can be accessed at https://atlas.fredhutch.org/kincyte. The source code and all other files are listed in the Key Resources Table. To contribute data, kindly email the corresponding author and consult *Supplementary file 2* for guidance on the preferred file format.

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
