## [Editor Report · eLife assessment]

This manuscript describes an **important** web resource for kinases connected to cytokines. The **compelling** information will be used by researchers across a number of fields including analysts, modelers, wet lab experimentalists and clinician-researchers, who are looking to improve our understanding of pathologies and means to correct them through modulating the immune response.

---

## [Referee Report · Reviewer #1 (Public Review)]

Summary:

Kinase inhibitors represent a highly valuable class of drugs as evidenced by their continued clinical success. The target landscape of kinase targeting small molecules can be leveraged to alter multiple phenotypes with increasing complexity that broadly aligns with increasing target promiscuity. This 'tools and resources' contribution provides a starting point for researchers interested in aligning kinase inhibitor activity with cytokine/chemokine stimulated signal transduction networks.

Strengths:

KinCytE is a forward-thinking database that yields hypothesis-generating options for researchers interested in pharmacologically modulating cytokine/chemokine signaling.

Weaknesses:

As a 'tools and resources' contribution, the primary (potential) weakness will be the authors' willingness to update and improve the tool. KinCytE will require frequent updating to better inform users in terms of contextual cytokine/chemokine stimulated signaling and the target landscape of those agents that are included as options.

---

## [Referee Report · Reviewer #2 (Public Review)]

Summary:

In this manuscript, "KinCytE- a Kinase to Cytokine Explorer to Identify Molecular Regulators and Potential Therapeutic", the authors present a web resource, KinCytE, that lets researchers search for kinase inhibitors that have been shown to affect cytokine and chemokine release and signaling networks. I think it's a valuable resource that has a lot of potential and could be very useful in deciding on statistical analysis that might precede lab experiments.

Opportunities:

With the release of the manuscript and the code base in place, I hope the authors continue to build upon the platform, perhaps by increasing the number of cell types that are probed (beyond macrophages). Additionally, when new drug-response data becomes available, perhaps it can be used to further validate the findings. Overall, I see this as a great project that can evolve.

Strengths:

The site contains valuable content, and the structure is such that growing that content should be possible.

Weaknesses:

Only based on macrophage experiments, would be nice to have other cell types investigated, but I'm sure that will be remedied with some time.

---

## [Author Response]

The following is the authors’ response to the original reviews.

We thank the editor and the reviewers for their valuable and constructive feedback. In the revised manuscript, we have incorporated and addressed the suggestions provided by the reviewers.

**Reviewer #1 (Recommendations For The Authors):**
The primary recommendation is to provide additional language explaining how KinCytE will be updated.

Response: We appreciate the reviewer’s insightful feedback regarding the KinCytE update. In response, we have included additional details in the “Development and use of KinCyte’ section as follows: “We welcome researchers to actively participate in advancing the development of KinCytE by sharing external screening data, especially data on new secreted factors and cell types that extend beyond macrophages. This collaborative effort promises to enhance our understanding of kinase-focused networks, opening new avenues for cutting-edge therapeutic approaches”. In addition, we explicitly state in the "Data, Software, and Availability" section, "To contribute data, kindly email the corresponding author and refer to Table S2 for guidance on the preferred file format."

**Reviewer #2 (Recommendations For The Authors):**
Would have been nice to see a validation of the regression models from outside of the training data. I would also consider removing statements like "We anticipate that KinCytE will be highly sought after by biologists... " , it reads like a grant application (and this is not)! Could tone the language down a bit. In the future, you might consider displaying your graphs as "biofabrics", they're much cleaner than "hairballs" (PMID: 23102059). Or potentially, show a hierarchical view where the selected cytokine (or other) is at the root, and you can immediately see what's connected. Anyway, the network display can be expanded. Consider maybe adding the nearest neighbors to the table on the right after selecting the node. Generally, though, I like how it works.There needs to be a button to download the graph as a .csv file. Maybe the subgraph after selecting a node (or set of nodes). Also, once you're at a graph view, it's hard to guess how to get back to the starting page. Maybe just one button with a "home" on it would fix that. On the Kinases Discovery, why are the gene symbols all lower case? Very cool!

Response:: We greatly value the reviewer's constructive suggestions. To incorporate these, we have made the following changes:

(1) "We anticipate that KinCytE will be highly sought after by biologists... " This sentence is removed.

(2) A ‘SAVE CSV’ button is added to the bottom right of the Cytokine Explorer page, which allows the users to download the graph as a csv file.

(3) A redesigned KinCyte logo now functions as the 'HOME' button, located at the top left of the webpage, ensuring that users can easily return to the homepage at any time.